# Nuclear division phenotypes in Sporidiobolales and Trichosporonales

Keita Aoki,[1] Moriya Ohkuma,[2] Takashi Sugita,[3] Yuuki Kobayashi,[1] Naoto Tanaka,[4] Masako Takashima[1]

**ABSTRACT**    During nuclear division in basidiomycetous yeasts, the mother cell nucleus migrates to the daughter cell and divides inside that cell, followed by nuclear segregation in both cells. This mode contrasts sharply with that of *Saccharomyces cerevisiae*, where nuclear division occurs within the mother cell without prior nuclear migration. However, the universality of this division mode across all basidiomycetous yeasts remains unclear. Accordingly, we aimed to identify basidiomycetous yeasts that undergo nuclear division inside their mother cells by observing nuclear division in 51 Sporidiobolales and 33 Trichosporonales yeasts. The phenotype of a budded cell with nuclei that exists only in the daughter cell is a key indicator of nuclear migration to the daughter cell prior to nuclear division. Using 4′,6-diamidino-2-phenylindole staining, we identified four candidate Trichosporonales species (*Apiotrichum domesticum*, *Cutaneotrichosporon arboriforme*, *Trichosporon asahii*, and *Vanrija humicola*) that exhibited a noticeable decrease in this phenotype. Conversely, all tested yeasts in Sporidiobolales showed the phenotype. Live cell imaging of *T. asahii* expressing an mCherry-tagged histone H2A protein confirmed nuclear division occurring inside the mother cell or at the bud neck, without prior nuclear migration to the daughter cell. Furthermore, we found that a nucleus is divided near the center of a cell in *T. asahii*. The specific location—mother or daughter cells—where nuclear division occurs depends on the length of the daughter cell in *T. asahii*. These findings provide insight into nuclear division mode diversity and may support a better understanding of fungal pathogenesis.

**IMPORTANCE**    The dogma of nuclear division in basidiomycetous yeasts is that the nucleus present in the mother cell moves to the daughter cell and divides inside that cell. The mode of nuclear division is completely different from that in ascomycetous yeasts. Our results indicate that nuclear division follows the dogma in Sporidiobolales but not in all Trichosporonales yeasts. In the study, we identified four candidate species that do not follow basidiomycetous nuclear division (basidio-nuclear division), then we focused on *Trichosporon asahii* for deeper analysis. *T. asahii* surprisingly undergoes nuclear division inside the mother cell. Furthermore, we determined that the exact position of the nuclear division depends on the daughter cell length. This is the first report to identify *T. asahii* as a basidiomycetous yeast that does not follow basidio-nuclear division.

**KEYWORDS**    nuclear division, Trichosporonales, Sporidiobolales, *Trichosporon asahii*, fungal dimorphism

The faithful inheritance of chromosomes by daughter cells is essential for cell proliferation. Nuclear segregation and cell division are orchestrated to avoid missegregation of chromosomes and cell death, even in lower eukaryotes such as yeasts (1). In *Saccharomyces cerevisiae*, a model budding yeast, early mitotic chromosomes with short spindle microtubules are produced between duplicated spindle pole bodies embedded in the nuclear envelope of the mother cell during the early M phase (2). In

Address correspondence to Keita Aoki, ka207755@nodai.ac.jp, or Masako Takashima, mt207623@nodai.ac.jp.

The authors declare no conflict of interest.

See the funding table on p. 15.

*S. cerevisiae*, cytoplasmic microtubules originating from microtubule-organizing centers embedded in the nuclear envelope radiate outward and interact with the cell cortex to orient the spindle across the mother-to-daughter bud axis (3, 4). Cytoplasmic microtubules interact with the septin ring present in the bud neck to align with the spindle in the bud neck (5). The nucleus present in the mother cell moves to the bud neck and segregates, followed by the migration of daughter chromosomes toward the mother and daughter cells during anaphase (4). Spatial regulation of mitotic chromosomes is also observed in the budding cells of *Candida albicans*, a dimorphic ascomycetous yeast (4). Microtubule-associated motor proteins play major roles in microtubule cytoskeleton assembly, nuclear movement, and chromosome segregation in *C. albicans* (6–8). When dynein is mutated, abnormal positioning of the nucleus and the accumulation of bi-nucleation are observed in *S. cerevisiae* and *C. albicans* (5, 6). Therefore, *S. cerevisiae* and *C. albicans* nuclei are accurately positioned at the bud neck via cytoskeletal networks ahead of segregation.

In *Schizosaccharomyces pombe*, a model fission yeast, the nucleus is an anchor for interphase microtubule-organizing centers and is positively placed at the center of a cell via a microtubule-pushing mechanism (9). The nucleus contains the Mid1 protein, which is a major factor in the division-plane definition (10). During the G2/M phase, Mid1 is exported from the nucleus and accumulates at the medial cortex, thereby coupling the division plane and nuclear positions (10–13). Therefore, mitotic chromosomes align at the center of the long axis of rod-shaped cells during early mitosis and segregate into two daughter cells of equal size during anaphase, followed by septum formation.

In basidiomycetous yeasts, nuclear migration is regulated by the cytoskeletal networks of cytoplasmic microtubules, microtubule-organizing centers, and dynein, which share striking similarities with those of ascomycetous yeasts (14–17). A notable characteristic of nuclear division in basidiomycetous yeasts is that the nucleus migrates to a daughter cell and undergoes mitotic division. Subsequently, one of the two nuclei migrates back into the mother cell. This phenotype has been reported in *Cryptococcus neoformans* (15), *Ustilago maydis* (16), and *Cryptococcus laurentii* (18). Contact between cytoplasmic microtubules and the cell cortex aids the nucleus in entering daughter cells in *Cr. neoformans* (19, 20) and *U. maydis* (21). In *Cr. neoformans*, a nucleus is transported to the daughter cell via cytoplasmic microtubules by a pulling force generated by microtubule-binding proteins Bim1 and dynein, which are localized to the daughter cell cortex (17). However, nuclear division has been underexplored in *Trichosporon asahii*, a dimorphic yeast that causes deep-seated mycosis in immunocompromised patients (22, 23). Nuclear migration transiently creates dikaryotic daughter cells and anucleate mother cells. This phenotype is a hallmark of the nucleus migrating into a daughter cell ahead of mitotic division in budded cells and is not observed in ascomycetous yeasts. The detailed mechanism and biological significance of nuclear migration in basidiomycetous yeast remain unclear (4).

Nuclear division, a characteristic feature in basidiomycetes and termed basidio-nuclear division in this study, has been elucidated in only a limited number of basidiomycetous species. Therefore, this study aimed to examine whether basidio-nuclear division is common among basidiomycetous yeasts and identify basidiomycetous yeasts that do not follow basidio-nuclear division by observing nuclear division phenotypes in 51 Sporidiobolales and 33 Trichosporonales species based on 4′,6-diamidino-2-phenylindole (DAPI) staining and live cell imaging.

## RESULTS

### Nuclear division phenotypes in Sporidiobolales and Trichosporonales yeasts

The basidio-nuclear division cycle established in basidiomycetous yeasts was classified into four phenotypes according to nuclear positions: (i) budded cell harboring a nucleus only in the mother, (ii) budded cell harboring a nucleus only in the daughter, (iii) budded cell harboring divided nuclei only in the daughter or around the bud neck; (iv) budded cell with nuclear segregation across the bud neck, and (v) budded cell harboring one

nucleus each (Fig. 1A). The phenotype of a budded cell with its nuclei located only in the daughter is characteristic of basidio-nuclear division (Fig. 1A and B). Using DAPI staining, we observed nuclear division phenotypes in 51 Sporidiobolales and 33 Trichosporonales yeasts to identify basidiomycetous yeasts that do not follow basidio-nuclear division (Tables 1 to 3). These species were classified phylogenetically as basidiomycetes (Fig. S1). In Sporidiobolales, budding yeast cells, but not hyphal cells, were observed in all tested species. Among the Sporidiobolales species, the budded cells harboring nuclei only in the daughter were observed in all tested species (Fig. 2A; Table S1). Furthermore, divided nuclei were observed in the daughter cell in the following 14 species: *Rhodosporidiobolus microsporus*, *Rhodosporidiobolus poonsookiae*, *Rhodosporidiobolus ruineniae*, *Rhodotorula babjevae*, *Rhodotorula diobovata*, *Rhodotorula kratochvilovae*, *Rhodotorula taiwanensis*, *Rhodotorula toruloides*, *Sporobolomyces blumeae*, *Sporobolomyces johnsonii*, *Sporobolomyces phaffii*, *Sporobolomyces roseus*, *Sporobolomyces ruberrimus*, and *Sporobolomyces shibatanus* (Fig. 2A and B). In the rest of the Sporidiobolales species, nuclear division was observed near the bud neck (Fig. 2A and C). These phenotypes were different from *S. cerevisiae*, in which budded cells harboring nuclei only in their daughters were not observed (Fig. 2A and B). Early mitotic chromosomes were observed in the mother but not in the daughter of *S. cerevisiae* (Fig. 2B). The frequency of budded cells harboring nuclei only in the daughter was <1% only in *Rhodotorula mucilaginosa* (Fig. 2A; Table S1). In 33 Trichosporonales yeasts, the hyphal and fission forms were dominant in 13 species, and budding yeast cells were observed in the remaining 20 species (Table S1). Among the Trichosporonales yeasts, *Takashimella tepidaria* and *Takashimella koratensis*

**A**

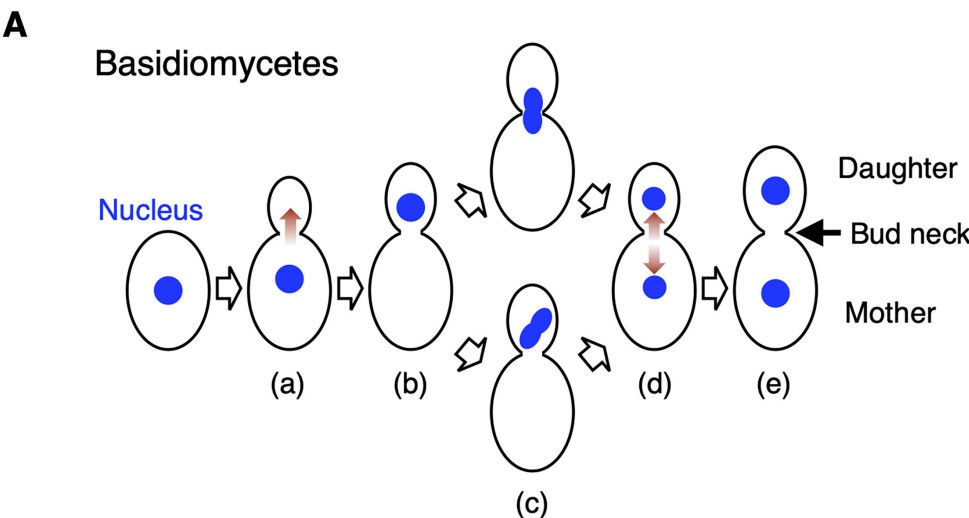

**B**

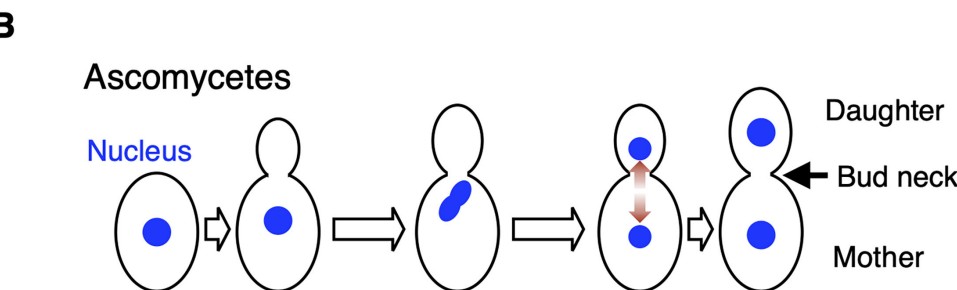

FIG 1 Nuclear division models in ascomycetous and basidiomycetous yeasts. (A) A schematic model representing basidio-nuclear division in basidiomycetous yeasts (18). (a) A nucleus present in a mother cell migrates to a daughter cell in the direction of a red arrow. (b) A nucleus is located only in a daughter cell. (c) A nucleus divided in a daughter cell or around a bud neck. (d) One nucleus moves back to a mother cell. (e) Both mother and daughter cells include one nucleus. (B) A schematic model representing nuclear division in ascomycetous yeasts.

**TABLE 1** Strains used in the study

| Species | Strains |
|---|---|
| *Trichosporon asahii* | JCM 2466 |
| *Trichosporon asteroids* | JCM 2937 |
| *Trichosporon coremiiforme* | JCM 2938 |
| *Trichosporon faecale* | JCM 2941 |
| *Trichosporon inkin* | JCM 9195 |
| *Trichosporon ovoides* | JCM 9940 |
| *Apiotrichum brassicae* | JCM 1599 |
| *Apiotrichum domesticum* | JCM 9580 |
| *Apiotrichum gamsii* | JCM 9941 |
| *Apiotrichum gracile* | JCM 10018 |
| *Apiotrichum laibachii* | JCM 2947 |
| *Apiotrichum montevideense* | JCM 9937 |
| *Apiotrichum porosum* | JCM 1458 |
| *Apiotrichum veenhuisii* | JCM 10691 |
| *Cutaneotrichosporon arboriforme* | JCM 14201 |
| *Cutaneotrichosporon cavernicola* | HIS 02 |
| *Cutaneotrichosporon cavernicola* | HIS 19 |
| *Cutaneotrichosporon cavernicola* | HIS 631 |
| *Cutaneotrichosporon cavernicola* | HIS 641 |
| *Cutaneotrichosporon cavernicola* | HIS 712 |
| *Cutaneotrichosporon aff. cavernicola* | HIS 471 |
| *Cutaneotrichosporon curvatum* | JCM 1532 |
| *Cutaneotrichosporon cutaneum* | JCM 1462 |
| *Cutaneotrichosporon cyanovorans* | JCM 31833 |
| *Cutaneotrichosporon daszewskae* | JCM 11166 |
| *Cutaneotrichosporon dermatis* | JCM 11170 |
| *Cutaneotrichosporon mucoides* | JCM 9939 |
| *Cutaneotrichosporon spelunceum* | HIS 16 |
| *Takashimella koratensis* | JCM 12878 |
| *Takashimella tepidaria* | JCM 11965 |
| *Pascua guehoae* | JCM 10690 |
| *Prillingera fragicola* | JCM 1530 |
| *Vanrija humicola* | JCM 1457 |
| *Saccharomyces cerevisiae* | BY 1438 |

exhibited the phenotype of budded cells with nuclei only in the daughter in 8.7% and 4.2% of all budded cells, respectively, and showed divided nuclei in the daughter cell (Fig. 3A and B). In the 20 species of Trichosporonales yeasts, the frequency of budded cells harboring nuclei only in the daughter was <1% in the following eight species: *Apiotrichum domesticum*, *Cutaneotrichosporon arboriforme*, *Cutaneotrichosporon cyanovorans*, *Cutaneotrichosporon daszewskae*, *Cutaneotrichosporon dermatis*, *T. asahii*, *Trichosporon inkin*, and *Vanrija humicola* (Fig. 3A). Notably, the phenotype of budded cells harboring nuclei only in the daughter severely decreased in *A. domesticum*, *C. arboriforme*, *T. asahii*, and *V. humicola* (Fig. 3A). In *C. arboriforme* and *V. humicola*, an early mitotic chromosome was observed inside the mother cell, similar to that in *S. cerevisiae* (Fig. 3C). In *T. asahii*, budded cells harboring a nucleus only in the daughter appeared at a rate of 0.5% of all budded cells. This frequency was 1/17th of that observed in *Ta. tepidaria* (Fig. 3A). Nuclear division occurred in the mother cell, and one nucleus moved to the daughter cell (Fig. 3C), which are characteristic of *T. asahii* but not of *Ta. tepidaria*. Therefore, the phenotype of budded cells with nuclei only in the daughter was rare in some Trichosporonales yeasts including *T. asahii*.

To further examine the relationship between the frequency of budded cells harboring nuclei only in the daughter and the cell length of the daughter cell, we measured

**TABLE 2** Strains used in the study

| Species | Strains |
| --- | --- |
| *Rhodosporidiobolus azoricus* | JCM 11251 |
| *Rhodosporidiobolus fluvialis* | JCM 10311 |
| *Rhodosporidiobolus lusitaniae* | JCM 8547 |
| *Rhodosporidiobolus microsporus* | JCM 6882 |
| *Rhodosporidiobolus nylandii* | JCM 10213 |
| *Rhodosporidiobolus odoratus* | JCM 11641 |
| *Rhodosporidiobolus poonsookiae* | JCM 10207 |
| *Rhodosporidiobolus ruineniae* | JCM 1839 |
| *Rhodosporidiobolus ruineniae* | JCM 8097 |
| *Rhodotorula alborubescens* | JCM 5352 |
| *Rhodotorula araucariae* | JCM 3770 |
| *Rhodotorula babjevae* | JCM 9279 |
| *Rhodotorula dairenensis* | JCM 3774 |
| *Rhodotorula diobovata* | JCM 3786 |
| *Rhodotorula diobovata* | JCM 3787 |
| *Rhodotorula glutinis* | JCM 8208 |
| *Rhodotorula graminis* | JCM 3775 |
| *Rhodotorula kratochvilovae* | JCM 8171 |
| *Rhodotorula kratochvilovae* | JCM 8172 |
| *Rhodotorula mucilaginosa* | JCM 8115 |
| *Rhodotorula pacifica* | JCM 10908 |
| *Rhodotorula paludigena* | JCM 10292 |
| *Rhodotorula paludigena* | JCM 10293 |
| *Rhodotorula sphaerocarpa* | JCM 8202 |
| *Rhodotorula sphaerocarpa* | JCM 3791 |
| *Rhodotorula sphaerocarpa* | JCM 9055 |
| *Rhodotorula taiwanensis* | JCM 3773 |
| *Rhodotorula toruloides* | JCM 10021 |
| *Rhodotorula toruloides* | JCM 10020 |
| *Rhodotorula toruloides* | JCM 10022 |
| *Rhodotorula toruloides* | JCM 10049 |
| *Rhodotorula toruloides* | JCM 10297 |
| *Rhodotorula toruloides* | JCM 10295 |
| *Rhodotorula toruloides* | JCM 10298 |

daughter and mother cell lengths and calculated the ratio of daughter cell length to mother cell length in Trichosporonales yeasts (Fig. 3A). Since *Cutaneotrichosporon cavernicola* includes five strains and one affinity strain, it was a suitable species to compare frequencies of nuclear phenotypes in an intraspecies context. In *C. cavernicola* strains HIS 19 and HIS 712, an increase in budded cells with nuclei only in the daughter corresponded to a higher ratio of daughter to mother cell length (Fig. 3A). Therefore, we presumed that daughter cell length is related to the appearance of the phenotype of budded cells with nuclei only in the daughter.

## Construction of mCherry-fused histone H2A in *T. asahii*

To examine nuclear division in live *T. asahii* cells, we constructed recombinant strains expressing mCherry-tagged histone H2A. *T. asahii* genes annotated as histone H2A in the National Center for Biotechnology Information (NCBI) database (https://www.ncbi.nlm.nih.gov/) were A1Q1_00160, A1Q1_05283, and A1Q1_06997. The protein sequences of A1Q1_00160, A1Q1_05283, and A1Q1_06997 showed 60.36%, 71.67%, and 81.08% identity with that of YDR225W, which codes histone H2A gene on the genome in *S. cerevisiae*, respectively. The protein sequences of A1Q1_06997 showed

**TABLE 3** Strains used in the study

| Species | Strains |
|---|---|
| *Rhodotorula toruloides* | JCM 24501 |
| *Rhodotorula toruloides* | NBRC 10075 |
| *Rhodotorula toruloides* | NBRC 10076 |
| *Rhodotorula toruloides* | NBRC 10512 |
| *Rhodotorula toruloides* | NBRC 10513 |
| *Sporobolomyces salmonicolor* | JCM 1841 |
| *Sporobolomyces salmonicolor* | JCM 8246 |
| *Sporobolomyces salmonicolor* | JCM 21990 |
| *Sporobolomyces blumeae* | JCM 10212 |
| *Sporobolomyces carnicolor* | JCM 3766 |
| *Sporobolomyces johnsonii* | JCM 1840 |
| *Sporobolomyces koalae* | JCM 15063 |
| *Sporobolomyces patagonicus* | JCM 16287 |
| *Sporobolomyces phaffii* | JCM 11491 |
| *Sporobolomyces roseus* | JCM 5353 |
| *Sporobolomyces ruberrimus* | JCM 16303 |
| *Sporobolomyces shibatanus* | JCM 3765 |

81.4% identity with orf19.6924 (*Candida albicans*) and 89.15% identity with CNAG_06747 (*Cr. neoformans*) (Fig. 4A and B). Therefore, A1Q1_06997 seemed to be the most suitable ortholog of histone H2A, although the $NH_2$ and COOH terminal sequences were diverse among species (Fig. 4A). Therefore, we used A1Q1_06997 to obtain live images of nuclear division in *T. asahii*. We constructed an mCherry-fused histone H2A recombinant strain with a nourseothricin-resistant gene using MPU129 Δku70 (24) as a host strain (Fig. 4C). The insertion of mCherry-fused H2A into the genome was confirmed by colony polymerase chain reaction (PCR) (Fig. 4D) and microscopic observation (Fig. 4E). DAPI-stained signals co-localized with H2A-mCherry signals in fixed *T. asahii* cells (Fig. 4E). Consequently, the H2A-mCherry signal served as a nuclear localization marker in *T. asahii*.

## Live imaging of nuclear division in *T. asahii*

To examine how *T. asahii* cells undergo nuclear division, we performed time-lapse imaging of H2A-mCherry signals in Sabouraud medium supplemented with a yeast nitrogen base and observed nuclear division in 11 live cells. In 2 out of the 11 cells, we succeeded in observing the budding phenotype from the early growth of a bud to separate H2A-mCherry signals over 220 min (Fig. 5A). The trajectories of the H2A-mCherry signals based on the position of the bud neck were observed for 220 min (Fig. 5B). A budded daughter cell grew at the velocity of 0.049 ± 0.010 µm/min ($n = 2$) until nuclear division, although the size of the mother cell remains unchanged (Fig. 5A; Table S1). In 6 of the 11 cells, we directly observed the dynamics of H2A-mCherry signals. H2A-mCherry signals inside a mother cell moved closer to a bud neck at the velocity of 0.061 ± 0.038 µm/min ($n = 6$) and separated within 5 min at the velocity of 0.91 ± 0.40 µm/min ($n = 6$) near a bud neck (Fig. 5A through C). Therefore, nuclear division occurs inside the mother cell or around the bud neck, and one nucleus moves to a daughter cell in *T. asahii*.

Furthermore, to examine positions of the nucleus and the bud neck 5 min before nuclear division, relative positions were calculated. Five minutes before nuclear division, relative positions of the nucleus and the bud neck along the long axis against the edge of a mother cell were 0.49 ± 0.016 and 0.55 ± 0.11, respectively ($n = 11$) (Fig. 5D). Distances of the location of the nucleus and the bud neck from the center of a cell along the long axis were 0.22 ± 0.11 µm and 1.2 ± 1.1 µm, respectively ($n = 11$) (Fig. 5D). The distance of the location of the bud neck from the center of a cell was significantly

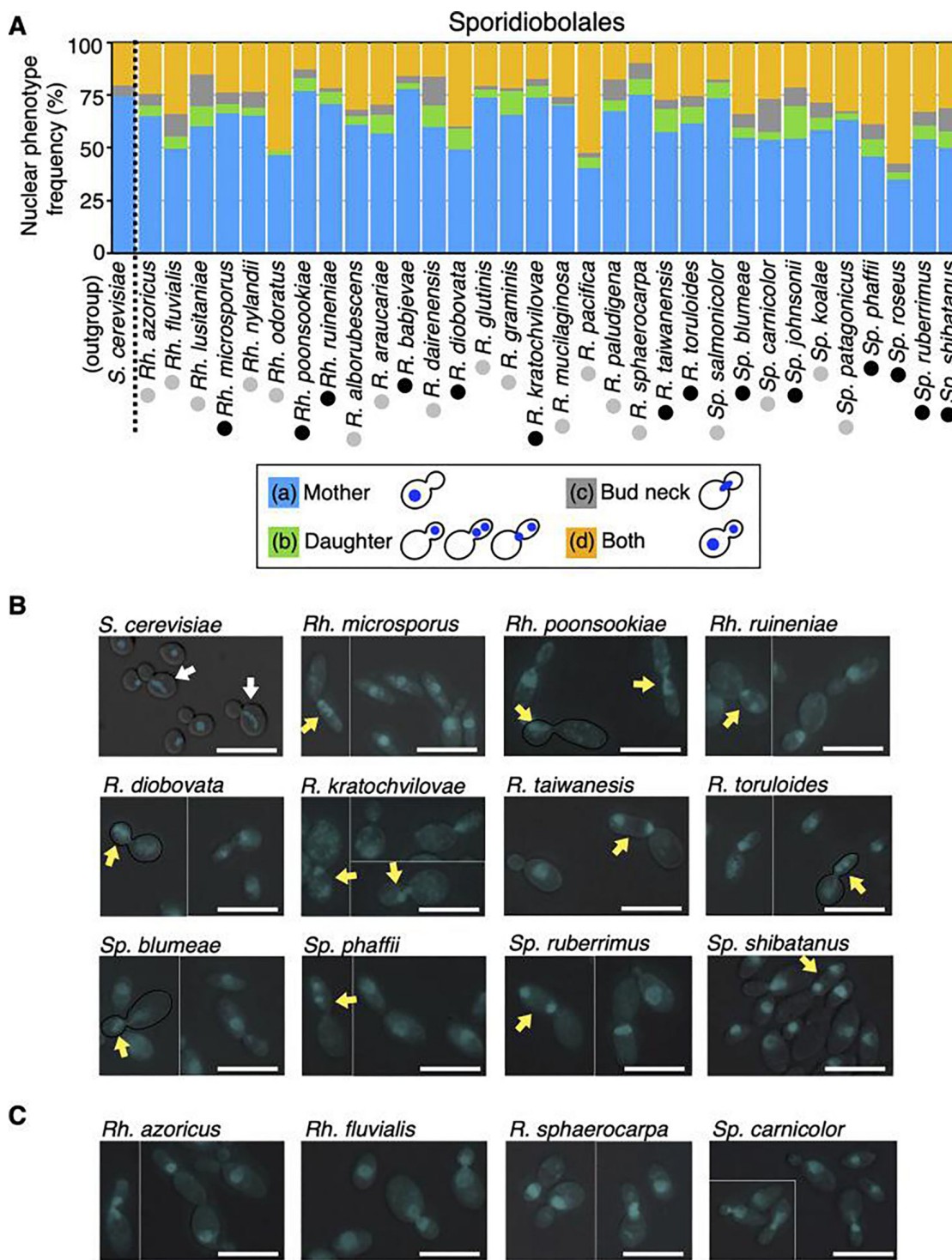

**FIG 2** Frequencies of nuclear division phenotypes in Sporidiobolales. (A) Nuclear phenotype frequencies were examined in 32 Sporidiobolales, with *S. cerevisiae* as a control. Nuclei were stained with 2.5 µg/mL DAPI in 70% ethanol-fixed cells. Budded cells were classified by nuclear positions that were only in the mother (blue bars), only in the daughter (green bars), around a bud neck (gray bars), and both in the daughter and mother (orange bars). Within the daughter-only category (green bars), black circles indicate species where divided nuclei or a mitotic nucleus was observed in the daughter cell, whereas the gray circles indicate species where these nuclei were not observed in the daughter cell. Abbreviations for the genera are as follows: *S*, *Saccharomyces*; *Rh*, *Rhodosporidiobolus*; *R*, *Rhodotorula*; *Sp*, *Sporobolomyces*. (B) DAPI phenotypes of yeasts whose nuclear division occurred in the daughter cell are shown. White arrows indicate mitotic nuclei located in the mother cells in *S. cerevisiae*. Yellow arrows indicate a nucleus or divided nuclei located only in the daughter cells. Black lines represent the external shapes of the cells. (C) DAPI phenotypes of yeasts whose nuclear division did not appear in the daughter cell are shown. DAPI, 4′,6-diamidino-2-phenylindole. Scale bar = 10 µm.

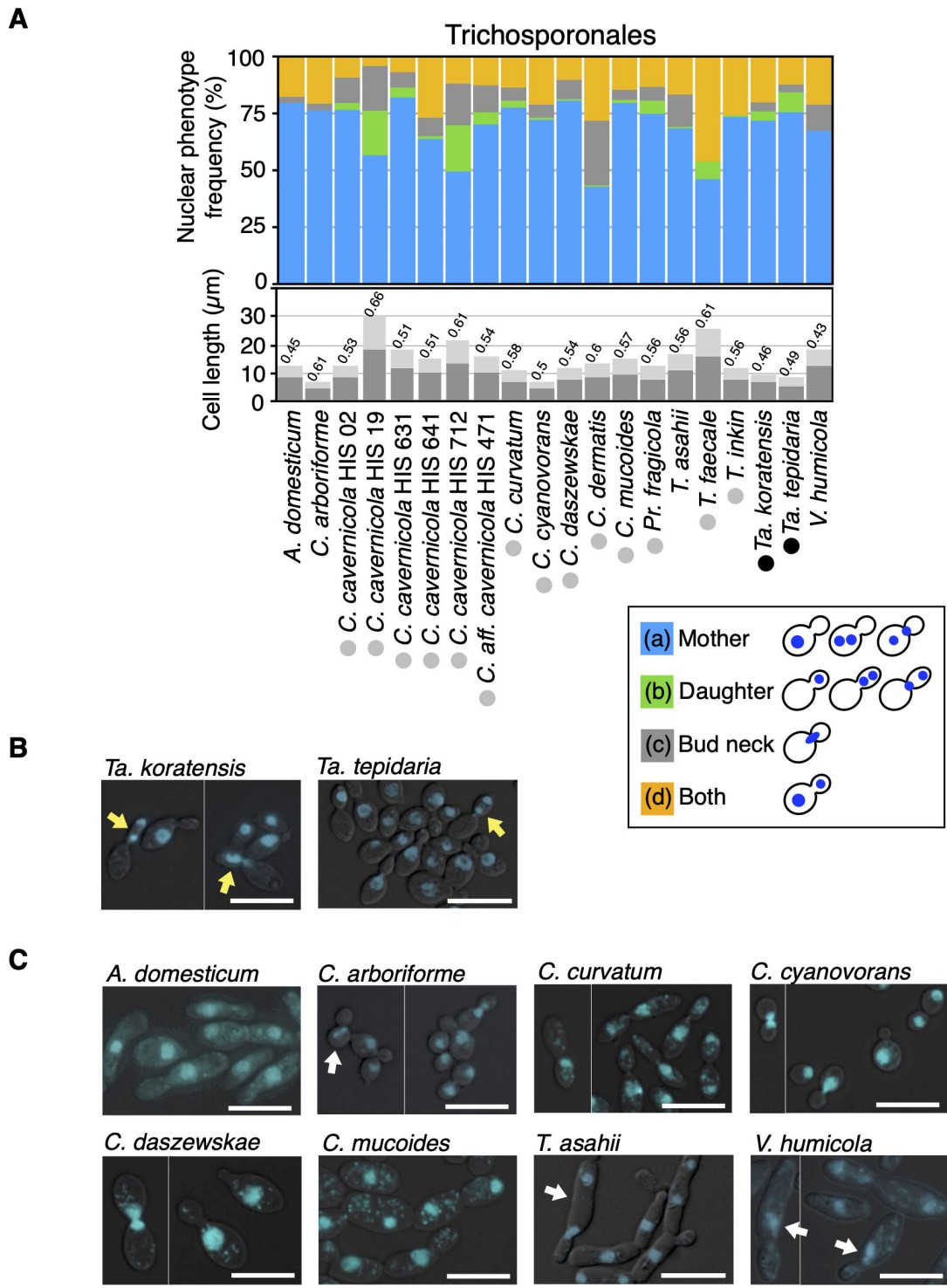

**FIG 3** Frequencies of nuclear division phenotypes in Trichosporonales. (A) Nuclear phenotype frequencies were examined in 20 Trichosporonales. Nuclei were stained with 2.5 µg/mL DAPI in 70% ethanol-fixed cells. Budded cells were classified by nuclear positions that were only in the mother (blue bars), only in the daughter (green bars), around a bud neck (gray bars), and both in the daughter and mother (orange bars). Within the daughter-only category (green bars), black circles indicate species where divided nuclei or a mitotic nucleus was observed in the daughter cell, whereas gray circles indicate species where these nuclei were not observed in the daughter cell. The average lengths of the mother and daughter cells are measured and shown by deep and light gray bars, respectively. The ratio of daughter cell length to mother cell length is represented above each bar. Abbreviations for the genera are as follows: *A, Apiotrichum*; *C, Cutaneotrichosporon*; *Pr, Prillingera*; *T, Trichosporon*; *Ta, Takashimella*; *V, Vanrija*. (B) DAPI phenotypes are shown in *Ta. koratensis* and *Ta. tepidaria*. Yellow arrows indicate a nucleus or divided nuclei located only in the daughter cells. (C) DAPI phenotypes of yeasts whose nuclear division did not appear in the daughter cell are shown. White arrows indicate mitotic nuclei located in the mother cells. DAPI, 4′,6-diamidino-2-phenylindole. Scale bar = 10 µm.

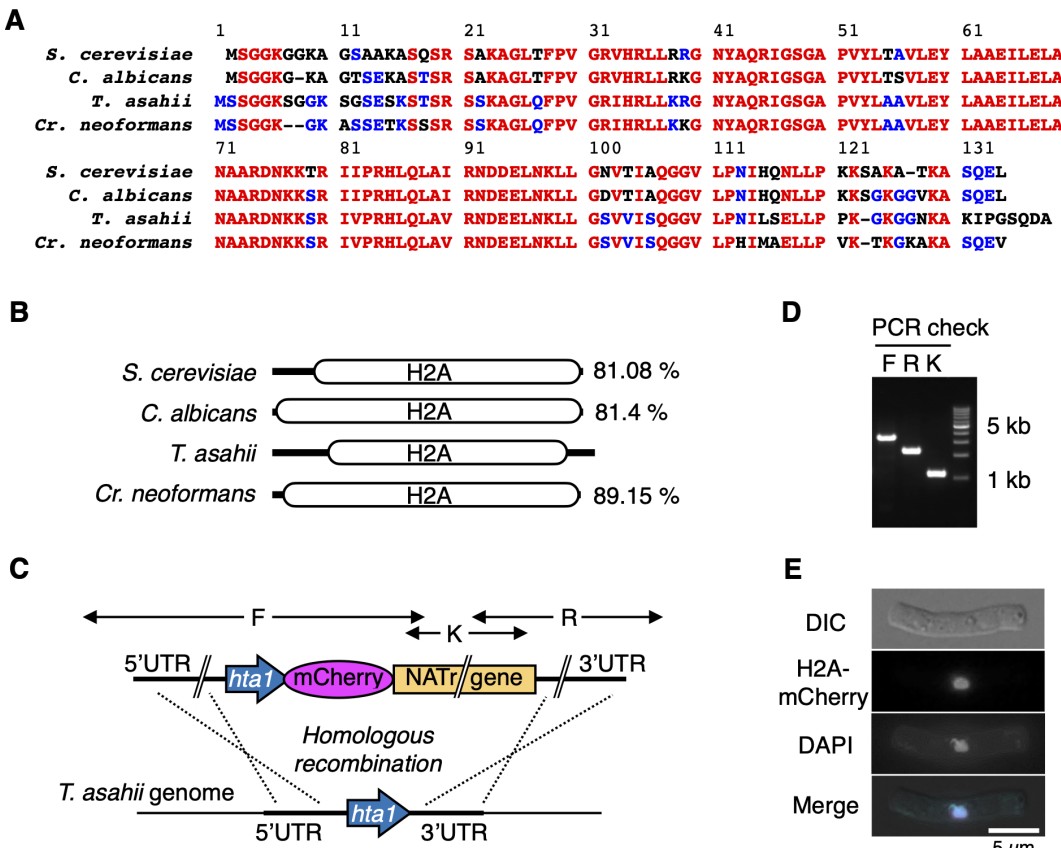

**FIG 4** Construction of mCherry-fused histone H2A in *T. asahii*. (A) Amino acid sequences of histone H2A were compared among *S. cerevisiae*, *C. albicans*, *T. asahii*, and *Cr. neoformans*. Red and blue residues indicate 100% and 50% identities among four yeasts, respectively. (B) A schematic model of histone H2A was compared among *S. cerevisiae*, *C. albicans*, *T. asahii*, and *Cr. neoformans*. Each identity against *T. asahii* H2A is shown. (C) A scheme about the replacement of the histone H2A gene with a mCherry-fused histone H2A gene on the *T. asahii* genome is described. (D) Results of colony PCR using primer sets F, R, and K are shown. (E) H2A-mCherry signals were merged with DAPI signals in cells that were fixed with 70% ethanol. Scale bar = 5 µm. UTR, untranslated region.

dispersed than that of the nucleus from the center of a cell (*P* = 0.01). Therefore, 5 min before nuclear division in *T. asahii*, the nuclei were consistently located near the center of a cell, whereas bud necks were not necessarily located near the center of a cell (*n* = 11).

## Putative orthologs involved in nuclear migration in *T. asahii*

To better understand the molecular basis of this nuclear division pattern, we performed a blastp search (https://blast.ncbi.nlm.nih.gov/Blast.cgi) for orthologs related to nuclear migration using the *S. cerevisiae* proteins as a query (Table 4). Orthologs in *S. cerevisiae* and *Cr. neoformans* were previously reported (25). Similar to *Cr. neoformans*, we found that *T. asahii* and *R. toruloides* lack Kar9 and She1 orthologs that are involved in nuclear migration and dynein inhibition, respectively (Table 4). However, unlike *Cr. neoformans*, *T. asahii* and *R. toruloides* lack Pac1 and Jnm1 orthologs but possess other dynactin complex components, such as Arp1 and Nip100 orthologs. Notably, the kinesin-5 orthologs Cin8 and Kip1 are encoded by a single, shared protein in both *T. asahii* and *R. toruloides*. Similarly, the Bik1 ortholog and a portion of the Nip100 ortholog are on the same single protein in *T. asahii* and *R. toruloides*.

## DISCUSSION

In this study, we observed nuclear division phenotypes in 51 Sporidiobolales and 33 Trichosporonales yeasts to identify species that do not follow basidio-nuclear division in basidiomycetes. In Sporidiobolales yeasts, we confirmed that nuclear division occurred

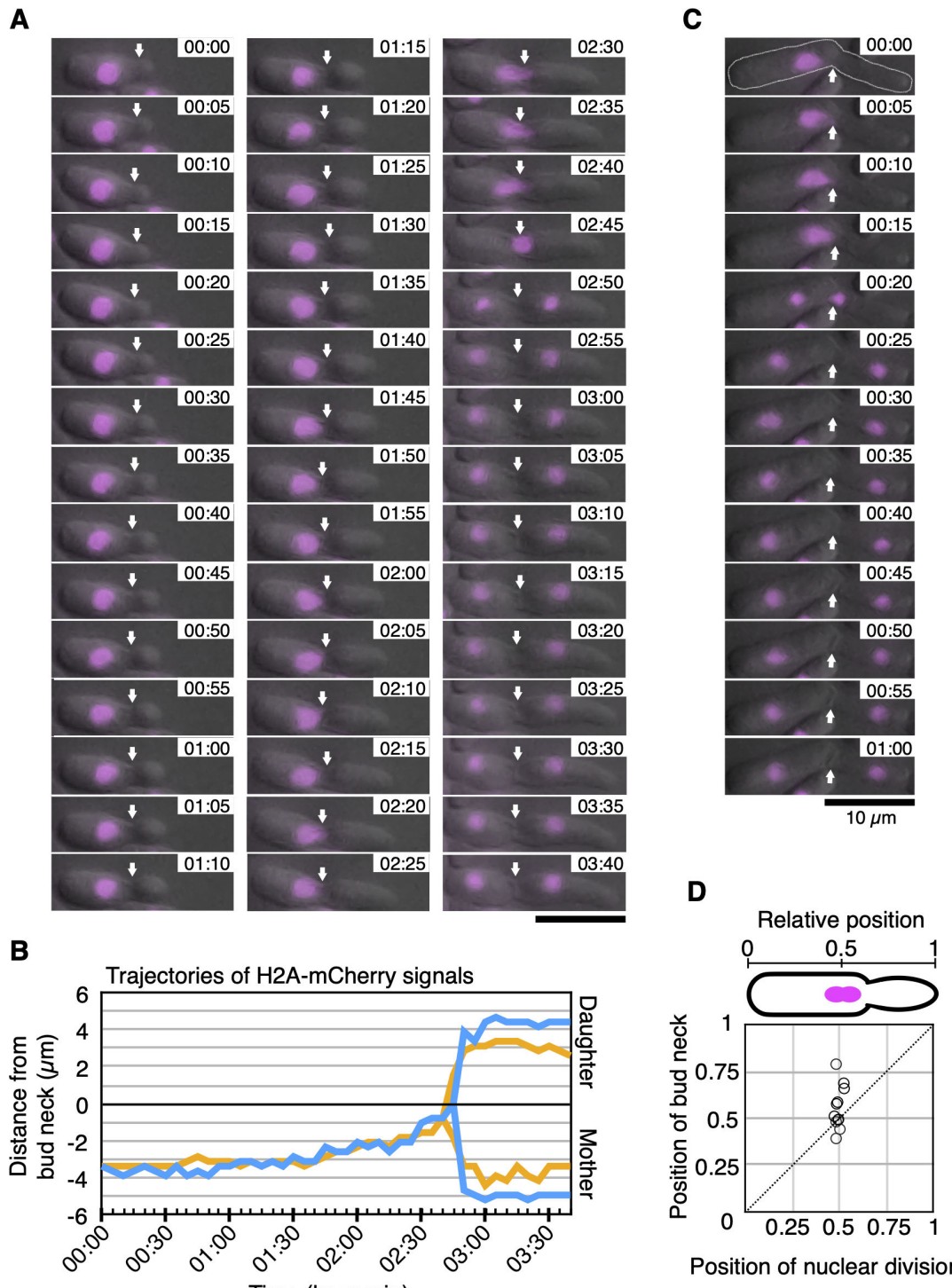

FIG 5 Live imaging of H2A-mCherry signals in *T. asahii* live imaging of H2A-mCherry signals was performed with 5 min intervals in *T. asahii* cells that were cultivated on Sabouraud agar medium supplemented with YNB at 25°C–28°C. H2A-mCherry signals are shown in magenta. (A) The H2A-mCherry signals divided at the bud neck. The arrows indicate the positions of the bud neck. Time indicates hour:minute. (B) Distances of H2A-mCherry signals from a bud neck at 0 µm are shown in two trajectories. The trajectory of the H2A-mCherry signals shown in panel A is shown in blue. (C) H2A-mCherry signals are divided inside the mother cells. The dotted line represents the cell shape. The arrows indicate the positions of the bud neck. (D) Relative positions of nuclear division and bud neck in 11 cells. Scale bar = 10 µm.

inside daughter cells, as reported previously (15, 16, 18), because budded cells harboring nuclei only in the daughter appeared in all the tested cells, whereas nuclear division did not occur inside daughter cells in all Trichosporonales yeasts because budded cells harboring nuclei only in the daughter rarely appeared in *A. domesticum*, *C. arboriforme*, *T. asahii*, and *V. humicola*. Therefore, these four species are candidates that do not undergo basidio-nuclear division. Furthermore, we showed that *T. asahii* does not undergo basidio-nuclear division in the study. We found a nucleus divided inside a mother cell without transportation to a daughter cell ahead of nuclear division in *T. asahii* using the time-lapse imaging method. We cannot perfectly deny the possibility of *T. asahii* nucleus moving from a mother cell to a daughter cell within 5 min; however, it is not realistic because the velocity of nuclear movement before nuclear division is $0.061 \pm 0.038$ μm/ min, such that a nucleus can only move 0.3 μm during 5 min. Therefore, it is conceivable that the position of nuclear division is not determined by the established mechanism, allowing for a new mechanism in nuclear division modes in *T. asahii*.

In time-lapse imaging using *T. asahii* cells, nuclei always divide near the center of the long axis of a cell, including both mother and daughter cells. The results support a novel hypothesis that a nucleus is necessarily divided at the center of a cell, and that the specific compartment—mother or daughter cells—where nuclear division occurs depends on the length of the two cells. In this hypothesis, when the mother cell length is longer than the daughter cell length, the nucleus divides inside the mother cell; conversely, when the mother cell length is shorter than the daughter cell length, the nucleus divides inside the daughter cell. As expected, high frequencies of budded cells harboring nuclei only in the daughter were accompanied by longer daughter cells in strains of HIS 19 and HIS 712 among the five strains of *C. cavernicola*, and a similar accompaniment of nuclear phenotype and daughter cell length was observed in *Trichosporon faecale*. However, all Trichosporonales yeasts do not always follow this hypothesis. In *C. arboriforme*, which does not exhibit the hyphal form, budded cells harboring nuclei only in the daughter did not appear, although the daughter cell length was relatively long among the Trichosporonales yeasts tested in the study. Therefore, there may be alternative modes of nuclear division, depending on whether dimorphism exists in Trichosporonales.

The mechanism of nuclear positioning and migration has been well studied in model budding yeasts across ascomycetes and basidiomycetes. In *S. cerevisiae* and *Cr. neoformans*, Bim1 and dynein are transmitted from the mother cell cortex to the daughter cell cortex as the daughter cell grows. Once localized, these proteins generate a pulling force on cytoplasmic microtubules, which in turn pulls the spindle toward the daughter cell (25). Since Bim1 and dynein orthologs are conserved in *T. asahii* and *R. toruloides* (Table 4), a similar pulling mechanism likely exists in these species. In *S. cerevisiae*, dynein movement along cytoplasmic microtubules is inhibited by She1 in a dynactin-independent manner (26). Furthermore, the contact between cytoplasmic microtubules and cell cortex is regulated by Kar9, and proteasome-mediated degradation of kar9 allows spindle alignment at the mother-daughter axis (27). Consistent with what has been reported for *Cr. neoformans* (25), She1 and Kar9 orthologs are also absent in *T. asahii* and *R. toruloides* (Table 4). The absence of these orthologs might promote nuclear movement toward the daughter cell. These data align with observations in *R. toruloides* but not with our findings for *T. asahii*. In *T. asahii*, nuclear division occurs near the center of a cell; therefore, some unknown mechanism must exist to limit the Bim1 and dynein activities. One possibility is that there is a functional Kar9 ortholog or a related protein that may exist in *T. asahii*, since six hypothetical proteins harboring the yeast cortical protein Kar9 (pfam08580) motif are annotated on the NCBI web site (https://www.ncbi.nlm.nih.gov/) in *T. asahii*. Another possibility is that the regulation of the cytoplasmic microtubule in *T. asahii* resembles that in the fission yeast *Schizosaccharomyces pombe*, which also lacks a Kar9 ortholog, as opposed to *S. cerevisiae*. The phenotype of nuclear division near the center of a cell is consistent with the pattern seen in *S. pombe*.

**TABLE 4** Key proteins involved in nuclear migration[a,b]

| S. cerevisiae | Cr. neoformans | T. asahii | R. toruloides | Function (25) |
|---|---|---|---|---|
| **Kinesins** | | | | |
| Cin8 | ND | A1Q1_07208 | RHTO_06593 | Kinesin-5 (+end) |
| Kar3 | CNAG_05752 | A1Q1_00311 | RHTO_04402 | Kinesin (−end) |
| Kip1 | CNAG_03453 | A1Q1_07208 | RHTO_06593 | Kinesin-5 (+end) |
| Kip2 | CNAG_07817 CNAG_06335 | A1Q1_06432 | RHTO_01024 | Centromere protein E |
| Kip3 | CNAG_00172 | A1Q1_05621 | RHTO_04728 | Kinesin-related motor |
| Smy1 | CNAG_06728 | A1Q1_05161 | RHTO_04402 | Interacts with Myo2 |
| **+TIPs** | | | | |
| Bik1 | CNAG_06352 | A1Q1_02081 | RHTO_00814 RHTO_06001 | MT + end-associated protein |
| Bim1 | CNAG_03993 | A1Q1_07553 | RHTO_05016 | Generate MT capture site at the cell cortex along with Kar9 |
| Kar9 | ND | ND | ND | Nuclear migration |
| Pac1 | CNAG_07440 | ND | ND | LIS1/NudF homolog Nuclear migration |
| **Dynein/Dynactin** | | | | |
| Dyn1 | CNAG_05894 | A1Q1_04433 | RHTO_07288 | Dynein heavy chain |
| Pac11 | CNAG_04407 | A1Q1_02759 | RHTO_02117 | Dynein intermediate |
| Arp1 | ND | A1Q1_08064 | RHTO_04841 | Dynactin complex |
| Arp10 | CNAG_04196 | A1Q1_06613 | ND | Dynactin complex |
| Jnm1 | CNAG_03966 | ND | ND | Dynactin complex |
| Nip100 | ND | A1Q1_02081 A1Q1_01754 | RHTO_01897 RHTO_00814 RHTO_06001 | Dynactin complex |
| **Regulators** | | | | |
| Ipl1 | CNAG_01285 | A1Q1_01256 | RHTO_07697 | Aurora kinase B |
| She1 | ND | ND | ND | Inhibition of dynein |

[a]Key proteins for nuclear migration (25) are listed. Orthologs in *S. cerevisiae* and *Cr. neoformans* were identified based on a previous report (25). Orthologs in *T. asahii* (strain CBS 2479) and *R. toruloides* (strain NP11) were identified using the blastp tool (https://blast.ncbi.nlm.nih.gov/Blast.cgi) using the *S. cerevisiae* proteins as query. ND indicates that no ortholog protein was detected.
[b]MT, microtubule.

 *A. domesticum*, *C. arboriforme*, *T. asahii*, and *V. humicola*, which rarely contain budded cells harboring nuclei only in their daughters, exist across four genera of Trichosporonales. Similar to ascomycetous yeasts, these yeasts are presumed to possibly undergo nuclear division within the mother cell. The mechanism that enables the diversification of the nuclear division position in Trichosporonales yeasts has not been investigated. One possibility is that the position of nuclear division is influenced by hyphal formation mechanisms. Previously, hyphal formation has been observed in *A. domesticum*, *T. asahii*, and *V. humicola* (28, 29). Furthermore, this phenotype may be characteristic of Trichosporonales yeasts because basidio-nuclear division occurs in *Cryptococcus* spp. (15, 18), *Ustilago maydis* (16), and Sporidiobolales genera (this study). Therefore, it was speculated that genetic diversification occurred in an ancestral yeast of Trichosporonales.

 We examined nuclear division phenotypes based on the budding mechanism in 84 species of Sporidiobolales and Trichosporonales. This comprehensive observation was the first trial in basidiomycetes and offered new insight into the position of nuclear division in Trichosporonales yeasts. Our results suggest that diverse modes of nuclear division exist in Trichosporonales but not in Sporidiobolales yeasts. To our knowledge, this study is the first to demonstrate that nuclear division in basidiomycetous

yeasts occurs within the mother cells. Nuclear division modes in dimorphic yeasts can potentially be differentiated from others in Trichosporonales yeasts. However, distinguishing budded cells in dimorphic yeasts is challenging owing to their large size and often unclear bud neck position. Therefore, future research should employ a protein marker for the bud neck to accurately determine the budding site in dimorphic yeasts. The study of nuclear division in Trichosporonales yeasts supports the understanding of dimorphism and pathogenicity of dimorphic yeasts.

## MATERIALS AND METHODS

### Strains and media

All strains used in the study are listed in Tables 1 to 3. The JCM and NBRC strains were provided by the Japan Collection of Microorganisms (RIKEN BioResource Research Center, Tsukuba, Japan) and the NITE Biological Resource Center (National Institute of Technology and Evaluation, Kisarazu, Japan), respectively. *S. cerevisiae* strain BY 1438 was provided by the National BioResource Project (NBRP) in Japan (JPNBRP202225). These strains were cultivated in yeast peptone dextrose (YPD) broth (2% glucose, 1% yeast extract, and 2% Bacto peptone) at 25°C before observations.

### Cell staining

One milliliter of cells grown to log phase was harvested and fixed with 1 mL of 70% ethanol for 1 h at 4°C. The cells were washed twice with 1 mL phosphate-buffered saline (PBS) and suspended in 50 µL PBS. One microliter of the cell mixture was mixed using 1 µL of 5 µg/mL DAPI solution on a slide glass before being covered with a cover glass for the observations. Cells were observed under an OLYMPUS BX53 microscope at ×40 or ×100 magnification using a U-FDICT mirror unit (Olympus, Tokyo, Japan). U-FMCHE and U-FUNA mirror units were used to observe the mCherry and DAPI signals, respectively (Olympus). The light was supplied using a U-LGPS system (Olympus).

### Alignment of histone H2A proteins

The protein sequences of *S. cerevisiae* YDR225W, *C. albicans* orf19.6924, *T. asahii* A1Q1_06997, and *Cr. neoformans* CNAG_06747 were obtained from the NCBI database (https://www.ncbi.nlm.nih.gov/) and aligned by using a sequence alignment server (http://multalin.toulouse.inra.fr/multalin/multalin.html).

### Construction of recombinant proteins

The 2 kb sequence, including A1Q1_06997 and 5′-untranslated region (UTR) (H2A-5) was amplified from the *T. asahii* genome using two primers: H2A-5F (5′-CTCGTCGTCTGAGG CCAGTGAGTC-3′) and H2A-5R (5′-GTTATCCTCCTCGCCCTTGCTCACAGCGTCCTGCGAGCCG GGGATCTT-3′). The H2A-5R primer included an $NH_2$-terminal 24 bp of mCherry DNA sequence. The DNA sequence of mCherry was amplified from BYP8912 (NBRP Yeast, Hiroshima, Japan) using two primers: mCherry-F (5′-GTGAGCAAGGGCGAGGAGGATAAC -3′) and mCherry-R (5′-AGGGTATTCTGGGCCTCCATGTCGTTACTTGTACAGCTCGTCCATGCC -3′). mCherry-R primer includes $NH_2$-terminal 24 bp of 5′-untranslated region upstream of the nourseothricin-resistant gene. The nourseothricin-resistant gene including 5′- and 3′-untranslated region (NATr) was amplified from FYP4244 (NBRP Yeast) using two primers: Nat-F (5′-CGACATGGAGGCCCAGAATACCCT-3′) and Nat-R (5′-CAGTATAGCGACC AGCATTCACAT-3′). The 2 kb sequence of 3′-UTR downstream of A1Q1_06997 (H2A-3) was amplified from *T. asahii* genome using two primers: H2A-3F (5′-ATGTGAATGCTGG TCGCTATACTGCCACCCGCCAATCTTGCCTTCCTC-3′) and H2A-3R (5′-CTTCGTCACCGAGACT GAGAACGA-3′). H2A-3F primer includes the $NH_2$-terminal 24 bp of the 3′-untranslated region downstream of A1Q1_06997. The combined DNA fragments, including H2A-5,

mCherry, NATr, and H2A-3, were amplified using H2A-5 and H2A-3R primers, resulting in the H2A-mCherry fragment used for transformation.

## Transformation

A modified version of the transformation method described by Matsumoto et al. (30) was used. MPU129 Δku70 (24, 31) cells were cultivated on Sabouraud agar medium (4% glucose, 1% hipolypeptone, and 2% agar) with 50 µg/mL G418 (Enzo, New York, USA) for 24 h at 25℃. The cells were suspended in 2 mL PBS on a Sabouraud agar medium, and the suspension was filtered through a 40 µm pore size cell strainer (AS ONE Corporation, Osaka, Japan). The 0.15 mL cell suspension after the filtration was plated on Sabouraud agar media and cultivated for 18 h at 25℃. The cells grown on a Sabouraud agar medium were suspended in 2 mL PBS again, filtered through a 40 µm pore size cell strainer, and transferred into a 1.5 mL tube. The cells were washed four times with ice-cold sterile water and suspended in 50 µL 1 M sorbitol. The cell suspension was mixed with H2A-mCherry fragment of 180 ng, and the cell mixture stood for 15 min at 4℃. The cell mixture was poured into a cuvette at a 0.2 cm interval and pulsed at 1.8 kV for 5 ms using a Gene Pulser Xcell (BioRad, California, USA). The cell mixture was immediately mixed with 0.5 mL YPD broth supplemented with 0.6 M sorbitol and cultivated for 3 h at 25℃. The cell culture was centrifuged at $9,060 \times g$ for 5 min at 20℃, and the supernatant was removed. The cell pellet was suspended with 100 µL PBS and plated onto a Sabouraud agar medium supplemented with 350 µg/mL nourseothricin (Jena Bioscience, Thuringia, Germany). To verify the replacement of the H2A gene with the mCherry-fused H2A gene in the T. asahii genome, colony PCR was performed by KOD One PCR enzyme (TOYOBO, Osaka, Japan) using three primer sets: primer set F (5′-CAGTCTGCAGGTCGACGATG-3′ and 5′-TAAGCCGTGTCGTCAAGAGT-3′), primer set R (5′- GCTCTACATGAGCATGCCCT-3′ and 5′-GACGTCGTTCACAGTGTCTC-3′), and primer set K (5′-CGACATGGAGGCCCAGAATACCCT -3′ and 5′-CAGTATAGCGACCAGCATTCACAT-3′) and resulted in 3.6, 2.4, and 1.1 kb DNA fragments, respectively.

## Live cell imaging

A 0.6 mL Sabouraud medium with 2% agar containing yeast nitrogen base was placed in a microscope glass slide with removable two-well chambers (Matsunami, Osaka, Japan), and the center of the Sabouraud medium was cut into 5 mm squares for cell culture. Cells were inoculated into Sabouraud broth supplemented with 4.15 mM magnesium sulfate at a concentration of $OD_{660} = 0.1$, and 50 µL of the cell mixture was poured into the square chamber and covered with an 18 mm square cover glass (Matsunami). Observations were performed at 5 min intervals under an OLYMPUS BX53 microscope at ×40 or ×100 magnification with a U-FDICT mirror unit (Olympus). The exposure time for the H2A-mCherry signals was 20–50 ms. The velocities of daughter and mother cell growth were calculated from 45 to 170 min and from 0 to 170 min, respectively. The velocity of nuclear migration before division was calculated from 100 to 165 min. The velocity of nuclear division was calculated from 170 to 175 min. U-FMCHE was used to observe the mCherry signals (Olympus). The light was supplied using a U-LGPS system (Olympus).

## Statistical analysis

Statistical differences between the relative positions of nuclear division and the bud neck along the long axis against the edge of the mother cell were calculated using variance test.

## ACKNOWLEDGMENTS

This study was supported by the Institute for Fermentation, Osaka, Japan.

We thank Editage for the English language editing.

## AUTHOR AFFILIATIONS

[1]Laboratory of Yeast Systematics, Tokyo NODAI Research Institute, Tokyo University of Agriculture, Setagaya, Tokyo, Japan

[2]Japan Collection of Microorganisms, RIKEN BioResource Research Center, Tsukuba, Ibaraki, Japan

[3]Department of Microbiology, Meiji Pharmaceutical University, Kiyose, Tokyo, Japan

[4]Department of Molecular Microbiology, Faculty of Life Sciences, Tokyo University of Agriculture, Setagaya, Tokyo, Japan

## AUTHOR ORCIDs

Keita Aoki http://orcid.org/0000-0003-2079-4031
Takashi Sugita http://orcid.org/0000-0002-2127-5017
Yuuki Kobayashi http://orcid.org/0000-0003-1061-3639
Masako Takashima http://orcid.org/0000-0002-7686-8661

## FUNDING

| Funder | Grant(s) | Author(s) |
| --- | --- | --- |
| Institute for Fermentation, Osaka | K-2020-006 | Masako Takashima |

## AUTHOR CONTRIBUTIONS

Keita Aoki, Conceptualization, Data curation, Formal analysis, Investigation, Methodology, Visualization, Writing – original draft, Writing – review and editing | Moriya Ohkuma, Resources, Supervision, Writing – review and editing | Takashi Sugita, Supervision, Validation, Writing – review and editing | Yuuki Kobayashi, Validation, Writing – review and editing | Naoto Tanaka, Project administration, Supervision, Writing – review and editing | Masako Takashima, Funding acquisition, Project administration, Supervision, Writing – review and editing

## ADDITIONAL FILES

The following material is available online.

### Supplemental Material

**Supplemental material (Spectrum01327-25-s0001.pdf).** Fig. S1; Tables S1 and S2.

### Open Peer Review

**PEER REVIEW HISTORY (review-history.pdf).** An accounting of the reviewer comments and feedback.

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
