## [Reviewer comments · Microbiology Spectrum]

Microbiology Spectrum

Nuclear division phenotypes in Sporidiobolales and Trichosporonales

Keita Aoki, Moriya Ohkuma, Takashi Sugita, Yuuki Kobayashi, Naoto Tanaka, and Masako Takashima

Corresponding Author(s): Keita Aoki, Tokyo Nogyo Daigaku

Review Timeline:

Submission Date:	April 28, 2025
Editorial Decision:	June 20, 2025
Revision Received:	August 18, 2025
Accepted:	August 28, 2025

Editor: Miguel Penalva

Reviewer(s): The reviewers have opted to remain anonymous.

Transaction Report:

DOI: <https://doi.org/10.1128/spectrum.01327-25>

Re: Spectrum01327-25 (Diverse nuclear division patterns observed in Trichosporonales)

Dear Dr. Keita Aoki:

Thank you for the privilege of reviewing your work. Below you will find my comments, instructions from the Spectrum editorial office, and the reviewer comments.

First of all, I should like to apologise for the long period required to collect reviews from relevant colleagues. As you may see, both reviewers made substantial criticisms to your submission. One of them was concerned about the statistical analysis methods to present your series of data. The second made a fair comment by indicating that the "static" studies that you are submitting fall somehow short of depth, as you are using a "fixed screenshot" and you are not analyzing the possibility that the nuclei move back-and-forth depending on the phase of mitosis. I will be happy to give you the opportunity of submitting a revised manuscript if you are positive that you will be capable of addressing the concerns of the referees

Revision Guidelines

Sincerely,
Miguel Penalva
Editor
Microbiology Spectrum

Reviewer #1 (Comments for the Author):

In the budding yeast *Saccharomyces cerevisiae*, nuclear division occurs in the mother cell, and one of the nuclei migrates into the daughter cell. In contrast, in some basidiomycetous yeasts such as *Cryptococcus neoformans*, *C. laurentii*, and *Ustilago maydis*, the nucleus first migrates into the daughter cell, and then one of the divided nuclei returns to the mother cell. This pattern, known as the "basidio-nuclear division cycle," raises the question of whether a similar mechanism operates in other basidiomycetous yeasts. The authors investigated this in 51 Sporidiobolales and 33 Trichosporonales species and found that four Trichosporonales yeasts undergo nuclear division inside the mother cell.

There is a certain value in investigating the diversity of nuclear division phenotypes in diverse yeasts. Two major patterns appear to exist: one in which the nucleus migrates into the daughter cell before division, and another in which nuclear division occurs near the center of the entire cell structure. I would like to see a more thorough discussion suggesting mechanistic differences behind these patterns, focusing especially on what is known in the budding yeast about the genes involved. If possible, it would also be helpful to include an analysis of whether there are differences in the conservation of such genes. Additionally, the treatment of significant figures is problematic. In particular, it is inappropriate to present values that fall below the resolution limit of light microscopy.

Specific comments.

Could the authors include a phylogenetic tree indicating the evolutionary and taxonomic positions of the Sporidiobolales and Trichosporonales?

In *T. tepidaria*, budded cells harboring a nucleus only in the daughter appeared in 8.7% of the cells. In *T. asahii*, the same phenotype was observed in only 1.6% of the cells. As shown in Fig. 1B for *T. asahii*, two nuclei located at the mother cell and the bud neck. Such cells do not clearly fall into any defined category. Do you have representative images where the nucleus is located exclusively at the bud neck?

Cell length measurements in Table S1. Please express the cell length data using appropriate significant figures. Also, note that values below the resolution limit of light microscopy are not meaningful.

L166, Nuclear position definition. How is the "nuclear position" defined in relation to the shape and size of the nucleus? What is the level of precision or significance of the numerical values presented? Is it correct to understand that nuclear division occurs near the center of the mother-daughter cell complex?

L188, Significant figures. Please confirm whether the use of significant digits in this section is appropriate.

L204-211, Data visualization. Could the authors present this data using dot plots or box plots to better illustrate the distribution and variance?

L221-222, Nuclear division position in *T. asahii*. The authors mention that in *T. asahii*, nuclear division tends to occur near the center of the mother-daughter cell complex. Is this tendency also observed more broadly among other Trichosporonales yeasts?

Reviewer #2 (Comments for the Author):

The research conducted by Aoki and colleagues centres on the identification of the mechanisms employed by specific yeast-like fungi to facilitate nuclear division, with a particular focus on distinguishing those that occur within the nucleus of the mother cell from those that are expected to undergo mitosis in the daughter cell. The work appears to be arduous, and more than 60 species of yeast-like fungi have been studied using DAPI staining. The study is intriguing; however, the methodology employed significantly restricts comprehension of the subject under investigation, which incorporates a substantial dynamic element. Consequently, the utilisation of fixed cells constitutes a substantial impediment.

The authors identify four types of cell stages in fixed cells. Understanding that they only use cells that are budding, they identify those that have a nucleus in the mother cell, those that have it in the daughter cell, those that observe the nucleus in the neck, and those that have two nuclei, one in each cell. Certainly, the most interesting fact is to determine the presence of a nucleus in the daughter cell. The high frequency of this case is indicative of the permanence of the mitotic nucleus in that compartment over time. However, all this work loses value because the process is not determined from a dynamic point of view. In other words, it is not demonstrated that mitosis, especially metaphase and anaphase, takes place in the daughter cell. It is possible that the nucleus moves back to the neck or to the mother cell, where most of the mitosis takes place.

It would be ideal to show the microtubules emanating from the spindle pole bodies (SPB), and whether there is a lag between SPB division between species. The location of the SPB in the longitudinal orientation to the cell division axis could be the cause of migration of the nucleus to the daughter cell. The subsequent division of the SPB in that compartment could lead to the nucleus returning to the mother cell compartment or remaining centered in the neck, simply by the opposed pulling forces derived from the new microtubules emanated from this daughter SPB. This scenario is not defined or considered in the work

carried out and is crucial to understanding the process of mitosis in these fungi. Simply observing the nuclei is a very limited way of studying this biological and cellular problem.

August 19, 2025

Prof. Miguel Penalva

Editor

Microbiology Spectrum

Dear Prof. Penalva,

These are our point-by-point responses to the comments and suggestions raised by Reviewer 1 and Reviewer 2. Bold-type text indicates the comments from the Reviewers.

To Reviewer 1

In the budding yeast *Saccharomyces cerevisiae*, nuclear division occurs in the mother cell, and one of the nuclei migrates into the daughter cell. In contrast, in some basidiomycetous yeasts such as *Cryptococcus neoformans*, *C. laurentii*, and *Ustilago maydis*, the nucleus first migrates into the daughter cell, and then one of the divided nuclei returns to the mother cell. This pattern, known as the "basidio-nuclear division cycle," raises the question of whether a similar mechanism operates in other basidiomycetous yeasts. The authors investigated this in 51 Sporidiobolales and 33 Trichosporonales species and found that four Trichosporonales yeasts undergo nuclear division inside the mother cell.

- **Response:** We appreciate the reviewer's accurate summary of our study background. To clarify our findings, the DAPI-staining experiment identified four Trichosporonales yeasts (*Apiotrichum domesticum*, *Cutaneotrichosporon arboriforme*, *Trichosporon asahii*, and *Vanrija humicola*) as candidates of the species that do not follow basidio-nuclear division. We focused on *Trichosporon asahii* as our primary example. Using live-cell time-lapsed imaging using H2A-mCherry, we confirmed that *T. asahii* undergoes nuclear division within the mother cell. To make this distinction clearer, we have reorganized the manuscript to present the DAPI staining as a screening method for finding candidate species, followed by definitive live-cell imaging of *T. asahii*.

There is a certain value in investigating the diversity of nuclear division phenotypes in diverse yeasts. Two major patterns appear to exist: one in which the nucleus

migrates into the daughter cell before division, and another in which nuclear division occurs near the center of the entire cell structure. I would like to see a more thorough discussion suggesting mechanistic differences behind these patterns, focusing especially on what is known in the budding yeast about the genes involved. If possible, it would also be helpful to include an analysis of whether there are differences in the conservation of such genes.

- **Response:** We agree with the reviewer that a deeper discussion of the mechanistic aspects of the observed phenotype is crucial to enrich the manuscript. To this end, we have compiled a list of proteins involved in nuclear migration based on Varshney et al. 2019 (doi: 10.1371/journal.pgen.1007959) and the new information is now included in Table 4. We chose to investigate the proteins rather than the genes, as suggested by the reviewer, since the proteins are the real effectors, which carry on the function. Orthologous proteins in *T. asahii* and *R. toruloides* were identified via the blastp tool (<https://blast.ncbi.nlm.nih.gov/Blast.cgi>) using the *S. cerevisiae* proteins as a query.

In *S. cerevisiae* and *Cr. neoformans*, Bim1 and dynein are transferred from the mother cell cortex to the daughter cell cortex as the daughter cell grows. Bim1 and dynein when localized enough on the daughter cell cortex they exert tension via cytoplasmic microtubules and pull the spindle toward the daughter cell. Since Bim1 and dynein orthologs are conserved in *T. asahii* and *R. toruloides*, therefore a similar mechanism most probably would work and pull the spindle toward the daughter cell. In *S. cerevisiae*, dynein movement along cytoplasmic microtubules is inhibited by She1 in a dynactin-independent manner. Furthermore, the contact between cytoplasmic microtubules and cell cortex is regulated by the yeast spindle positioning factor Kar9, and proteasome-mediated degradation of kar9 allows spindle alignment at the mother-daughter axis (Schweiggert et al. 2016 doi: 10.1016/j.devcel.2016.01.011). As well as *Cr. neoformans* reported in Varshney et al., She1 and Kar9 orthologs were absent in *T. asahii* and *R. toruloides* (Table 4). The absence of these orthologs might promote nuclear movement toward the daughter cell. These data align with the observation in *R. toruloides* but not in *T. asahii* in the current study. In *T. asahii*, nuclear division occurs near the center of a cell, therefore, some mechanism to limit the activities of Bim and dynein is currently unidentified. One possibility is that there is functional Kar9 ortholog or Kar9-related protein(s) in *T. asahii*

that would carry this function. For instance, there are six hypothetical proteins harboring a motif of the yeast cortical protein Kar9 (pfam08580) are annotated on the NCBI web site (<https://www.ncbi.nlm.nih.gov/>) in *T. asahii* proteome. Alternatively, the regulation of the cytoplasmic microtubule in *T. asahii* could resemble that in the fission yeast *Schizosaccharomyces pombe*, which does not harbor a Kar9 ortholog, as opposed to *S. cerevisiae*. A phenotype of nuclear division near the center of a cell corresponds with the case of *S. pombe*. These discussion points are now included in the modified manuscript (page 13 line 11). Furthermore, a brief introduction about Bim1 and dynein was added (page 5, line 13).

Additionally, the treatment of significant figures is problematic. In particular, it is inappropriate to present values that fall below the resolution limit of light microscopy.

- **Response:** We appreciate the reviewer's watchfulness regarding our data presentation. We have carefully reviewed and revised the numerical data in Table S1 to ensure significant figures and omitted any values that fall below the resolution limit of light microscopy.

Specific comments.

Could the authors include a phylogenetic tree indicating the evolutionary and taxonomic positions of the Sporidiobolales and Trichosporonales?

- **Response:** As requested, a phylogenetic tree illustrating the evolutionary relationship and taxonomic position of Sporidiobolales and Trichosporonales has been added in Fig. S1.

In *T. tepidaria*, budded cells harboring a nucleus only in the daughter appeared in 8.7% of the cells. In *T. asahii*, the same phenotype was observed in only 1.6% of the cells. As shown in Fig. 1B for *T. asahii*, two nuclei located at the mother cell and the bud neck. Such cells do not clearly fall into any defined category. Do you have representative images where the nucleus is located exclusively at the bud neck?

- **Response:** The reviewer's concern regarding the ambiguity of the category of the cells with nuclei at both the mother cell and the bud neck is well taken. In the revised manuscript, we have included an image of this specific phenotype in *T. asahii* (Fig. 3C). We hypothesize this phenotype occurs as the cell in the process of cell division one nucleus is moving toward the

mother cell periphery and the other is crossing the bud neck into the daughter cell. Since, such a phenotype is very rare, we categorized these cells in the DAPI staining analysis as ‘Mother’ in Fig. 3A because the nucleus has not entered the daughter cell.

Cell length measurements in Table S1. Please express the cell length data using appropriate significant figures. Also, note that values below the resolution limit of light microscopy are not meaningful.

- **Response:** All cell length data was revised in Table S1, expressing values with appropriate significance figures that are within the resolution limit of light microscopy.

L166, Nuclear position definition. How is the "nuclear position" defined in relation to the shape and size of the nucleus? What is the level of precision or significance of the numerical values presented? Is it correct to understand that nuclear division occurs near the center of the mother-daughter cell complex?

- **Response:** Nuclear position in the study is defined as the midpoint between the mother-side daughter-side edges of the nucleus, measured from the bud neck along the long axis of a cell. The shape of the H2A-mCherry signals is not largely altered through the nuclear division in Fig. 5A. The data in Fig. 5D indicate that the nuclei consistently are located near the center of the mother-daughter cell complex just before cell division, supporting the idea that nuclear division in *T. asahii* takes place in this central position.

L188, Significant figures. Please confirm whether the use of significant digits in this section is appropriate.

- **Response:** In the revised manuscript, average frequencies of nuclear division phenotypes in DAPI staining experiment were deleted. We no longer make a statistical argument about these frequencies, since we considered the DAPI staining a screening tool rather than a quantitative comparison one.

L204-211, Data visualization. Could the authors present this data using dot plots or box plots to better illustrate the distribution and variance?

- **Response:** In the revised manuscript, we have withdrawn the statistical argument based on the variance test for the DAPI staining results. The DAPI

staining experiment was used just for screening species that do not follow basidio-nuclear division. Therefore, the stacked bar graph remains the most effective way to visualize the results of DAPI screening. Accordingly, we kept the stacked bar format.

L221-222, Nuclear division position in *T. asahii*. The authors mention that in *T. asahii*, nuclear division tends to occur near the center of the mother-daughter cell complex. Is this tendency also observed more broadly among other Trichosporonales yeasts?

- **Response:** This is indeed an excellent and interesting question. We have not performed time-lapse experiments for other Trichosporonales yeasts in the current study. However, we can hypothesize that this central nuclear division phenotype is highly linked to hyphal formation, which is a characteristic of dimorphic yeasts such as *T. asahii*. We speculate that gene expression related to hyphal growth may advantageously position the nucleus at the center of the cell, supporting the growth of cell tips. We consider this topic an important area for future research.

To Reviewer 2

The research conducted by Aoki and colleagues centres on the identification of the mechanisms employed by specific yeast-like fungi to facilitate nuclear division, with a particular focus on distinguishing those that occur within the nucleus of the mother cell from those that are expected to undergo mitosis in the daughter cell. The work appears to be arduous, and more than 60 species of yeast-like fungi have been studied using DAPI staining. The study is intriguing; however, the methodology employed significantly restricts comprehension of the subject under investigation, which incorporates a substantial dynamic element. Consequently, the utilisation of fixed cells constitutes a substantial impediment.

The authors identify four types of cell stages in fixed cells. Understanding that they only use cells that are budding, they identify those that have a nucleus in the mother cell, those that have it in the daughter cell, those that observe the nucleus in the neck, and those that have two nuclei, one in each cell. Certainly, the most interesting fact is to determine the presence of a nucleus in the daughter cell. The high frequency of this case is indicative of the permanence of the mitotic nucleus in that compartment over time. However, all this work loses value because the process is not determined from a dynamic point of view. In other words, it is not demonstrated that mitosis, especially metaphase and anaphase, takes place in the daughter cell. It is possible that the nucleus moves back to the neck or to the mother cell, where most of the mitosis takes place.

- **Response:** We appreciate the reviewer's insightful comments and agree that dynamic imaging is critical for through comprehension of nuclear division. To address this concern in our revised manuscript, we have made a clearer distinction between our initial screening using the DAPI staining and our more detailed dynamic analysis.

Regarding the issue of DAPI staining losing value, we added a visual marker (black circles in Fig. 2A and 3A), to indicate which species showed actual divided nuclei or mitotic figures within the daughter cell. This confirms that in these species, mitosis takes place in the 'daughter' compartment. Nuclear divisions inside the daughter cell are shown in Fig. 2B and 3B. Divided nuclei or mitotic figures were observed inside the daughter cells in the following 16 species: *Rhodosporidiobolus microsporus*, *Rh. poonsookiae*, *Rh. ruineniae*, *Rhodotorula babjevae*, *R. diobovata*, *R.*

kratochvilovae, *R. taiwanesis*, *R. toruloides*, *Sporobolomyces blumeae*, *Sp. johnsonii*, *Sp. phaffii*, *Sp. roseus*, *Sp. ruberrimus*, *Sp. shibatanus* (Fig. 2B), *Takashimella tepidaria*, and *Ta. koratensis* (Fig. 3B). Text describing the changes were added in (page 7, line 15) and (page 8, line 3) in the modified manuscript.

On the other hand, in the rest of the species showing the phenotype of budded cell harboring nuclei only in the daughter cell, nucleus was located at the bud neck frequently, although divided nuclei and mitotic figures were neither observed in the mother cell nor in the daughter cell. Therefore, we acknowledge that, as reviewer 2 suggested, it is possible that a nucleus moves back to the bud neck, where mitosis takes place. These species were marked with a gray circle in Fig. 2A and 3A.

Furthermore, to address the reviewer's concern about the lack of dynamic data, we performed the time-lapse imaging of H2A-mCherry signals in *T. asahii*. This live imaging (now presented in Fig. 5) provides definitive evidence that nuclear division in *T. asahii* takes place in the central position of the mother-daughter cell complex. This is the central finding of our study and directly addresses the limitations of the fixed cell approach.

It would be ideal to show the microtubules emanating from the spindle pole bodies (SPB), and whether there is a lag between SPB division between species. The location of the SPB in the longitudinal orientation to the cell division axis could be the cause of migration of the nucleus to the daughter cell. The subsequent division of the SPB in that compartment could lead to the nucleus returning to the mother cell compartment or remaining centered in the neck, simply by the opposed pulling forces derived from the new microtubules emanated from this daughter SPB. This scenario is not defined or considered in the work carried out and is crucial to understanding the process of mitosis in these fungi. Simply observing the nuclei is a very limited way of studying this biological and cellular problem.

- **Response:** We agree with Reviewer 2 that a detailed investigation of microtubules and SPB dynamics would be an excellent follow up on the current study. The scenario proposed by the reviewer involving opposed pulling forces from SPBs is an intriguing hypothesis to be tested in the future. However, the primary objective of the current study was to find whether basidio-nuclear division pattern is universal across basidiomycetous yeasts.

By screening a large number of species then confirming our findings with live-cell imaging for *T. asahii*, we provided the evidence that the pattern is not universal and identified *T. asahii* as the first basidiomycetous yeast to have a new mode of nuclear division in basidiomycetous yeasts. We acknowledge that studying details of microtubules and SPB dynamics is crucial for the full understanding of this mechanism and is of interest for us in future works.

Re: Spectrum01327-25R1 (Nuclear division phenotypes in Sporidiobolales and Trichosporonales)

Dear Dr. Keita Aoki:

Your manuscript has been accepted, and I am forwarding it to the ASM production staff for publication. Your paper will first be checked to make sure all elements meet the technical requirements. ASM staff will contact you if anything needs to be revised before copyediting and production can begin. Otherwise, you will be notified when your proofs are ready to be viewed.

Sincerely,
Miguel Penalva
Editor
Microbiology Spectrum

Reviewer #1 (Comments for the Author):

The authors addressed all suggestions satisfactorily.

Reviewer #2 (Comments for the Author):

Authors have accomplished all my doubts and queries